# EVO-RAG: Evolving Retrieval-Augmented Agents for Efficient Multi-Hop Query Optimization

## Abstract

Retrieval-augmented generation (RAG) grounds large language models (LLMs) in external evidence, yet *multi-hop* pipelines still suffer from redundant sub-queries, shallow exploration, and premature or delayed stopping. We present EVO-RAG, a phase-aware framework that couples a lightweight two-stage curriculum (**Discovery→Refinement**) with **seven step-level rewards** and an **in-episode time scheduler**. The scheduler decays exploration incentives as evidence accumulates while increasing efficiency and correctness pressure as uncertainty shrinks. Beyond scalar rewards, we train a **multi-head preference model** and benchmark **DPO**, **PPO**, and **GRPO** under *identical* rollouts and curricula for a controlled comparison. Evaluated on HotpotQA, 2WikiMultiHopQA, MuSiQue, and Bamboogle with 8B-class backbones, EVO-RAG improves EM/F1 while reducing redundant hops. Ablations show that (i) suppressing query overlap, (ii) rewarding *controlled backtracking* and *justified refusal*, and (iii) time-dynamic weighting are key to the accuracy–efficiency trade-off.

## 1 Introduction

Large language models (LLMs) deliver strong results in QA, dialogue, and text generation Brown et al. (2020); Ouyang et al. (2022); Raffel et al. (2020), yet they still hallucinate when relying on static pretraining. Retrieval-Augmented Generation (RAG) grounds responses in external documents Lewis et al. (2020), but *multi-hop* QA remains difficult: an agent must issue a *sequence* of sub-queries, integrate partial evidence, and decide when to backtrack, answer, or refuse.

Modern RAG pipelines span query rewriting, retrieval, filtering/reranking, and answer generation Chen et al. (2024b); Gao et al. (2024). End-to-end objectives that couple retriever and generator reduce handoff errors Chen et al. (2024b); Gao et al. (2024); Xiong et al. (2025), but most supervision is *static* and phase-agnostic. As a result, systems often over-search early or fail to consolidate late. RL-based approaches attempt to align modules to task rewards, yet many depend on episode-level signals and fixed weight schedules, offering weak credit assignment for intermediate actions and poor guidance on the transition from *exploration* to *refinement* Huang et al. (2025); Song et al. (2025); Liu et al. (2025); Sun et al. (2025).

We introduce EVO-RAG, a phase-aware framework for multi-hop retrieval (Fig. 1). The agent operates in two stages—*Discovery* then *Refinement*—and receives seven interpretable *step-level* signals: retrieval hit/miss, retrieval-action penalty, sub-query overlap, backtrack, refusal validity, step cost, and answer correctness. A time-based scheduler adjusts signal weights *within each episode*, decaying exploration incentives as evidence accumulates while increasing efficiency and correctness pressure as uncertainty shrinks. Beyond scalar rewards, we build a multi-head preference model that scores trajectory prefixes along these aspects, enabling both preference alignment (DPO) and scalarized policy gradients (PPO/GRPO) under identical rollouts.

Our design couples two time scales. Across training, a lightweight two-stage curriculum sets *endpoints* for each reward weight (discovery→refinement) without freezing behavior; within episodes, linear interpolation by progress ratio $p(t)$ produces smooth, phase-aware guidance. This separation keeps implementation simple—no additional modules at inference time—while providing dense, interpretable feedback for control-flow decisions such as BACKTRACK and REFUSE.

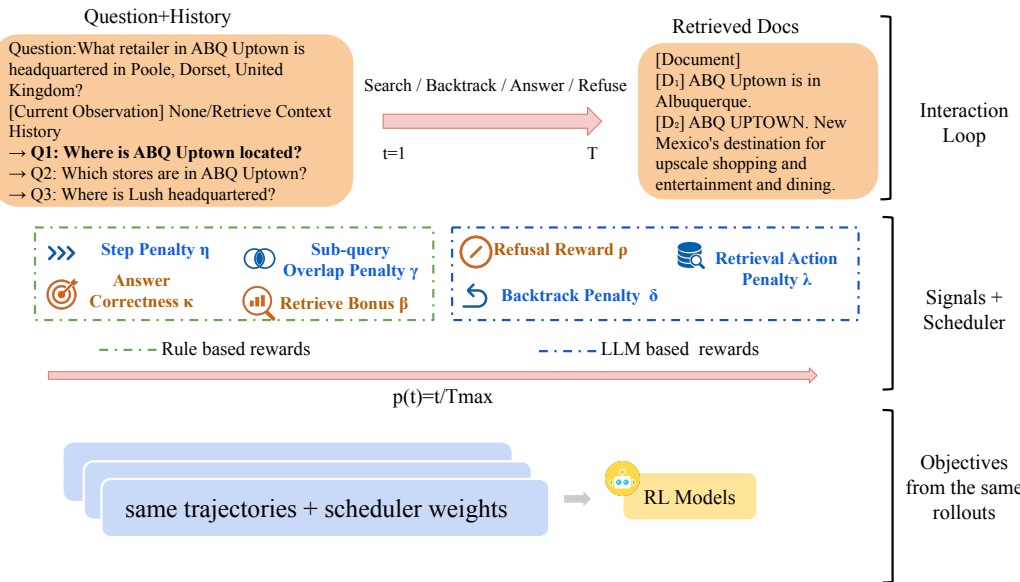

Figure 1: **EVO-RAG overview.** Left: at hop $t$ the agent rewrites a sub-query and retrieves evidence; four actions are available (continue, backtrack, answer, refuse). Right: seven step-level signals—*Retrieval Bonus*, *Retrieval Action Penalty*, *Sub-query Overlap Penalty*, *Backtrack Penalty*, *Refusal Reward*, *Step Penalty*, *Answer Correctness*—with a time-based scheduler that shifts emphasis from exploration to refinement. Lower: we train the same rollouts with three policy objectives (DPO/PPO/GRPO) for transparent per-dataset/per-backbone comparison.

We evaluate on HotpotQA, 2WikiMultiHopQA, MuSiQue, and Bamboogle Yang et al. (2018); Ho et al. (2020); Trivedi et al. (2022b); Press et al. (2022) using 8B-class backbones. To probe generalization, we train on a small HotpotQA subset and test across datasets. EVO-RAG improves EM/F1 while curbing redundant hops; ablations show that (i) suppressing query overlap, (ii) rewarding controlled backtracking and justified refusal, and (iii) time-based weighting are key to the accuracy–efficiency trade-off. We also report a controlled comparison of DPO, PPO, and GRPO on the *same* rollouts and curricula, surfacing objective-dependent differences without confounds. Our code is available at https://anonymous.4open.science/r/evorag-0C08/README.md..

## 2 RELATED WORK

**RAG paradigms.** RAG combines retrieval with generation to improve factuality and reduce hallucinations Lewis et al. (2020). Comparative studies map retriever–generator trade-offs and explore end-to-end/process supervision that tightens interaction between the two Chen et al. (2024b); Gao et al. (2024); Xiong et al. (2025). These objectives are typically *static* and phase-agnostic; our approach provides step-level, time-scheduled guidance within an episode to reflect evolving information needs in multi-hop reasoning.

**Query rewriting and multi-hop retrieval.** Multi-hop QA requires issuing well-formed sub-queries conditioned on partial evidence; errors propagate if later hops inherit poor queries. Methods mitigate this with missing-entity completion, interleaved reasoning–retrieval (e.g., IRCoT), speculative querying, and coherence-aware reranking Trivedi et al. (2022a); Wang et al. (2024); Zhang et al. (2024); Wei et al. (2024); Zhu et al. (2025). These approaches seldom supervise *when* to diversify versus consolidate. Our overlap and step-cost signals explicitly regulate duplication and chain length, while backtrack/refusal signals shape control flow.

**RL and preference optimization for RAG.** RL has been used to align retrieval, reranking, and generation (e.g., multi-agent or curriculum-based training) and to leverage preference objectives such as DPO/GRPO Chen et al. (2025); Huang et al. (2025); Ramesh et al. (2024); Kaiser & Weikum

(2025); Liu et al. (2025). Most rely on episode-level rewards with fixed schedules, which weakens credit assignment for intermediate actions. EVO-RAG instead supplies seven *step-level* signals with an in-episode scheduler and reports DPO/PPO/GRPO under identical rollouts for a controlled comparison.

## 3 METHOD

### 3.1 PRELIMINARIES: AGENTIC RAG ON RAG-GYM

We build on the high-level MDP abstraction of agentic RAG popularized by prior toolkits ("RAG-Gym"Xiong et al. (2025)). Each episode corresponds to one question $x$ and unfolds as a sequence $(s_t, a_t, o_t)_{t=1}^{T}$: **State** $s_t$: the question, the history of sub-queries and retrieved snippets, and the current scratchpad. **Observation** $o_t$: the top-$k$ passages returned by the IR system when $a_t =$ SEARCH, or $\varnothing$ for other actions. **Action space** $\mathcal{A}$: {SEARCH, BACKTRACK, ANSWER, REFUSE}. **Termination**: when the agent emits ANSWER or REFUSE, or $t = T_{\max}$.

This paper keeps the outcome reward on the final ANSWER (EM/F1) but *decomposes* process feedback at intermediate steps into seven interpretable signals (Sec. 3.3). A two-stage curriculum (Discovery→Refinement) and an in-episode time scheduler (Sec. 3.6) shape the relative weights of these signals.

### 3.2 REWARD SOURCES AND ACTION TRIGGERS

We use two feedback sources: (i) *rule-based* signals computable from the environment (hit gold doc, query redundancy, step cost), and (ii) *LLM-based* judgments for semantics-heavy cases (whether the current evidence suffices, thus REFUSE is justified). Table 1 summarizes *when* each signal fires and *who* provides it; formal definitions follow in Sec. 3.3.

Table 1: Process signals, trigger, and source. All symbols are defined in Sec. 3.3.

| Signal | Action / Timing | Source | Intuition |
|---|---|---|---|
| $r_{\text{ret}}$ (Retrieval Bonus) | SEARCH | Rule | reward early hits on $D^*$ |
| $r_{\text{dup}}$ (Overlap Penalty) | SEARCH | Rule | penalize redundant queries |
| $r_{\text{bt}}$ (Backtrack Pen.) | BACKTRACK | Rule | discourage blind backtracking |
| $r_{\text{ref}}$ (Refusal Reward) | REFUSE | LLM judge | refuse when evidence insufficient |
| $r_{\text{step}}$ (Step Cost) | every step | Rule | keep chains short |
| $r_{\text{act}}$ (Retrieval Act Pen.) | late SEARCH | Rule | curb late redundant searches |
| $r_{\text{ans}}$ (Answer Corr.) | terminal ANSWER | Rule | EM/F1 w.r.t. $A^*$ |

### 3.3 STEP-LEVEL REWARD

**Retrieval Bonus ($r_{\text{ret}}$).** At each step $t$, if the agent issues a SEARCH action that successfully retrieves any gold-supporting document $D^*$, it receives a positive reward; otherwise, a negative reward:

$$r_{\text{ret}}(s_t, a_t) = \begin{cases} +1 & a_t = \text{SEARCH} \wedge D_t \cap D^* \neq \varnothing, \\ -1 & a_t = \text{SEARCH} \wedge D_t \cap D^* = \varnothing, \\ 0 & \text{otherwise.} \end{cases}$$

This encourages early and effective retrieval.

**Sub-query Overlap Penalty ($r_{\text{dup}}$).** To discourage redundant sub-queries, we penalize cosine similarity between the current query $q_t$ and previous queries $q_j$:

$$r_{\text{dup}}(s_t, a_t) = -\max_{j<t} \cos(q_t, q_j).$$

**Backtrack Penalty ($r_{\text{bt}}$).** Whenever the policy selects BACKTRACK, we apply a fixed penalty:

$$r_{\text{bt}}(s_t, a_t) = -1[a_t = \text{BACKTRACK}].$$

**Refusal Reward ($r_{\text{ref}}$).** The agent is rewarded for refusing only when the retrieved evidence is insufficient, as verified by an external LLM:

$$r_{\text{ref}}(s_t, a_t) = \begin{cases} +1 & a_t = \text{REFUSE} \wedge \texttt{unanswerable}, \\ -1 & a_t = \text{REFUSE} \wedge \texttt{answerable}, \\ 0 & \text{otherwise.} \end{cases}$$

**Step Cost ($r_{\text{step}}$).** We discourage unnecessarily long reasoning chains:

$$r_{\text{step}}(s_t, a_t) = -1,$$

modulated by a dynamic weight $w_{\text{step}}(t)$ that increases with step count.

**Answer Correctness ($r_{\text{ans}}$).** At termination step $T$, correctness is measured by EM/F1 overlap with the ground-truth answer $A^*$:

$$r_{\text{ans}}(s_T, a_T) = \tfrac{1}{2}[EM(A_T, A^*) + F1(A_T, A^*)].$$

**Retrieval Action Penalty ($r_{\text{act}}$).** To limit late or redundant searches:

$$r_{\text{act}}(s_t, a_t) = \begin{cases} 0 & a_t = \text{SEARCH}, \ p(t) < 0.3, \\ -1[r_{\text{dup}} < 0] & a_t = \text{SEARCH}, \ p(t) \geq 0.3, \\ 0 & \text{otherwise.} \end{cases}$$

The total reward is an adaptive weighted sum $R_t = \sum_i w_i(t) r_i(s_t, a_t)$, with $w_i(t)$ annealed by the scheduler (see Section 3.5). This ensures different objectives dominate at appropriate reasoning phases.

## 3.4 PREFERENCE MODELING & POLICY OBJECTIVES

**Multi-head preference model.** Given rollouts with step-level labels $\{r_t^{(k)}\}_{k=1}^7$ and time weights $\{w_k(t)\}$, we construct preference pairs $(x^+, x^-)$ at the *trajectory-prefix* level using the weighted return $\sum_t \sum_k w_k(t) r_t^{(k)}$. A shared encoder with seven linear heads $\{f_\phi^{(k)}\}_{k=1}^7$ scores each aspect; the head-wise pairwise loss is

$$\mathcal{L}_{\text{RM}} = -\tfrac{1}{7} \sum_{k=1}^7 \log \sigma\big(f_\phi^{(k)}(x^+) - f_\phi^{(k)}(x^-)\big).$$

This factorization preserves interpretability and allows either preference- or reward-based policy learning.

**Path A: preference alignment (DPO).** We feed $(x^+, x^-)$ directly to the policy and optimize

$$\mathcal{L}_{\text{DPO}} = -\log \sigma\big(\beta_{\text{dpo}}[\log \pi_\theta(x^+) - \log \pi_\theta(x^-)]\big).$$

No scalarization is required.

**Path B: scalarized policy gradients (PPO / GRPO).** When desired, aspect scores (or environment labels) are linearly combined into a scalar step reward $\tilde{r}_t = \sum_k w_k(t) r_t^{(k)}$. We compute advantages with GAE and apply the PPO objective

$$\mathcal{L}_{\text{PPO}} = -\mathbb{E}\left[\min(r_t(\theta) A_t, \ \text{clip}(r_t(\theta), 1-\epsilon, 1+\epsilon) A_t)\right],$$

and its group-normalized variant (GRPO) by replacing $A_t$ with $\hat{A}_t^{(i)} = \frac{\tilde{r}_t^{(i)} - \mu_t}{\sigma_t + \varepsilon}$ over candidates.

**Objective summary.** We report all three objectives under identical rollouts and curricula. DPO consumes preferences; PPO/GRPO use the same per-step weights for scalarization. We refrain from universal claims; effects depend on dataset/backbone and are analyzed in Sec. 4.5.

Retrieval-focused weights $(\beta, \lambda)$ monotonically decrease, whereas efficiency-focused weights $(\gamma, \eta, \kappa)$ increase; the refusal weight $\rho$ stays constant. This "gearbox" provides step-level guidance that is missing from a static two-stage switch.

Table 2: Reward weights for EVO-RAG training. "Start" to "Mid" columns represent the interpolation range during Stage 1 (Discovery), and "Mid" to "End" represent Stage 2 (Refinement). Arrows ($\nearrow$, $\searrow$) indicate increasing or decreasing weight trends.

| Reward Component | Stage 1: Discovery | | | Stage 2: Refinement | | |
|---|---|---|---|---|---|---|
| | Start | Mid | Trend | Mid | End | Trend |
| Retrieval Bonus ($\beta$) | 2.0 | 1.0 | $\searrow$ | 1.0 | 0.5 | $\searrow$ |
| Retrieval Action Penalty ($\lambda$) | 1.5 | 0.8 | $\searrow$ | 0.8 | 0.4 | $\searrow$ |
| Sub-query Overlap Penalty ($\gamma$) | 0.1 | 0.5 | $\nearrow$ | 0.5 | 1.2 | $\nearrow$ |
| Backtrack Penalty ($\delta$) | 0.3 | 0.5 | $\nearrow$ | 0.5 | 1.0 | $\nearrow$ |
| Refusal Reward ($\rho$) | 0.5 | 0.5 | – | 0.5 | 0.5 | – |
| Step Penalty ($\eta$) | 0.02 | 0.05 | $\nearrow$ | 0.05 | 0.10 | $\nearrow$ |
| Answer Correctness ($\kappa$) | 0.05 | 0.10 | $\nearrow$ | 0.10 | 1.00 | $\nearrow$ |

### 3.5 TWO-STAGE CURRICULUM (ACROSS TRAINING)

We use two time scales for guidance: a *training-time* two-stage curriculum (Discovery $\rightarrow$ Refinement) and an *in-episode* time-based scheduler (Sec. 3.6). A "stage" does *not* fix weights; it only specifies the *interpolation endpoints*—Start$\rightarrow$Mid in Discovery and Mid$\rightarrow$End in Refinement—while the actual step-wise weights still evolve within each episode via $p(t)$.

**What each stage emphasizes.** **Discovery** exposes the policy to a high-entropy evidence space. We therefore set larger Start–Mid endpoints for retrieval-oriented terms ($w_\beta, w_\lambda$) and smaller ones for efficiency/precision ($w_\gamma, w_\eta, w_\kappa$) to encourage breadth and early hits. **Refinement** shifts these endpoints in the opposite direction (larger $w_\gamma, w_\eta, w_\kappa$, smaller $w_\beta, w_\lambda$), promoting consolidation, controlled stopping, and precise answering. The refusal weight $w_\rho$ stays constant across stages so that safe refusal is always available.

**When to switch stages.** We switch from Discovery to Refinement once exploration no longer increases the *composite return*. Concretely, let

$$R(\tau) = \sum_{t=1}^{T} \sum_k w_k(t)\, r_t^{(k)}$$

be the scalarized return of a trajectory $\tau$ on a held-out dev split (with $w_k(t)$ produced by the in-episode scheduler in Sec. 3.6). We keep an exponential moving average $\widehat{J}$ of episode-average returns and trigger the stage change when the improvement over the best running $\widehat{J}$ falls below a tolerance $\varepsilon$ for $P$ consecutive checkpoints (patience). Intuitively, once exploration plateaus, endpoints are shifted toward efficiency/accuracy for refinement.

**Why two stages rather than a fixed two-block schedule.** If one sets $w_k^{\text{early}} = w_k^{\text{late}}$ inside each stage, the scheme degenerates to a fixed two-block schedule. Our curriculum controls only the *endpoints*; the actual behavior in each episode is governed by the time-based scheduler below.

### 3.6 TIME-BASED SCHEDULER (WITHIN EPISODE)

**Progress ratio and interpolation.** We schedule step-level weights within each episode using a progress ratio

$$p(t) = \frac{t-1}{T_{\max}-1} \in [0,1],$$

where $t$ is the current step and $T_{\max}$ is the episode cap defined in §3.1. We then linearly interpolate between stage-specific endpoints ($w_k^{\text{early}}, w_k^{\text{late}}$):

$$w_k(t) = (1 - p(t))\, w_k^{\text{early}} + p(t)\, w_k^{\text{late}}.$$

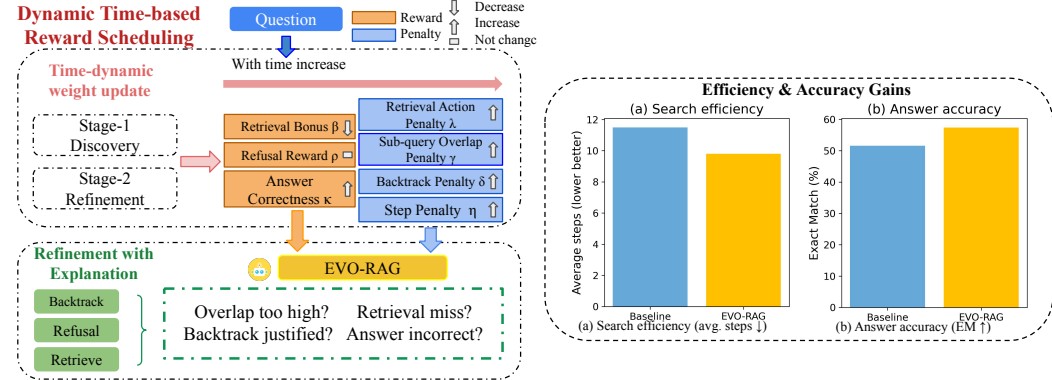

Figure 2: Time-based reward scheduling within an episode. Exploration weights $(\beta, \lambda)$ decay with progress $p(t)$, while efficiency/accuracy $(\gamma, \eta, \kappa)$ rise; refusal $(\rho)$ remains constant.

Table 3: Comparison of RAG methods on multi-hop QA datasets. Metrics are EM/F1 (stacked).

| Method | Backbone | HotpotQA (EM / F1) | 2Wiki (EM / F1) | MuSiQue (EM / F1) | Bamboogle (EM / F1) |
|---|---|---|---|---|---|
| RAG-Gym (ReSearch + PRM) | LLaMA-3.1-8B | 44.1 56.8 | 50.2 57.9 | 48.0 60.0 | **51.2** **63.1** |
| IRCoT (Flan-T5-XXL) | Flan-T5-XXL | 45.0 56.2 | 45.4 56.8 | 19.9 24.9 | 44.0 55.0 |
| **EVO-RAG** | DeepSeek-8B | **57.8** 71.4 | 52.6 66.4 | 51.8 63.7 | 45.3 58.2 |
| **EVO-RAG** | LLaMA-3.1-8B | 57.4 71.2 | 53.0 66.9 | 52.5 64.4 | 45.7 58.6 |
| **EVO-RAG** | Qwen-2.5-7B | 57.6 **71.5** | **53.2** **67.1** | 52.2 64.0 | 46.0 59.0 |

This guarantees $w_k(1) = w_k^{\text{early}}$ and $w_k(T_{\max}) = w_k^{\text{late}}$. If an episode terminates early (ANSWER/REFUSE), the schedule stops at the current $t$. We choose endpoints such that exploration-oriented $(w_\beta, w_\lambda)$ decrease, efficiency/accuracy $(w_\gamma, w_\eta, w_\kappa)$ increase, and $w_\rho$ remains constant. By choosing endpoints such that

$$w_\beta^{\text{early}} \geq w_\beta^{\text{late}}, \quad w_\lambda^{\text{early}} \geq w_\lambda^{\text{late}}, \qquad w_\gamma^{\text{early}} \leq w_\gamma^{\text{late}}, \quad w_\eta^{\text{early}} \leq w_\eta^{\text{late}}, \quad w_\kappa^{\text{early}} \leq w_\kappa^{\text{late}},$$

and $w_\rho^{\text{early}} = w_\rho^{\text{late}}$, we ensure exploration incentives $(w_\beta, w_\lambda)$ *decay* as evidence accumulates, while efficiency/accuracy $(w_\gamma, w_\eta, w_\kappa)$ *increase*; $w_\rho$ stays flat.

**Rationale.** (i) *Uncertainty reduction:* early steps face high-entropy evidence; rewarding early hits (large $w_\beta$) is valuable, but the marginal utility of additional searches diminishes with $p(t)$, so $w_\beta$ decays. (ii) *Cost–benefit dynamics:* late searches incur growing costs (latency, duplication, context pollution), hence we gradually strengthen overlap/step penalties $(w_\gamma, w_\eta)$ and answer accuracy $(w_\kappa)$. (iii) *Credit assignment:* terminal-only rewards poorly supervise BACKTRACK/REFUSE/STOP; reweighted step signals provide phase-appropriate gradients within the same episode.

**Implementation notes.** We use linear interpolation for reproducibility; other smooth monotone maps (e.g., sigmoid) are drop-in replacements. Boundary conditions are $w_k(1) = w_k^{\text{early}}$ and $w_k(T_{\max}) = w_k^{\text{late}}$.

Table 4: HotpotQA results under different reward schedules.

| Backbone | Strategy | EM | F1 |
|---|---|---|---|
| DeepSeek-8B | No Reward | 52.6 | 66.2 |
| | Two-stage | 55.0 | 68.7 |
| | **Time-dynamic** | **56.8** | **70.5** |
| LLaMA-3.1-8B | No Reward | 52.9 | 66.6 |
| | Two-stage | **57.4** | **71.2** |
| | **Time-dynamic** | 55.6 | 69.4 |
| Qwen2.5-7B-Instruct | No Reward | 53.1 | 66.7 |
| | Two-stage | 55.9 | 69.5 |
| | **Time-dynamic** | **57.6** | **71.5** |

Table 5: Single-Reward Ablation Results

| Single Reward Type | Eval Accuracy (%) | Eval Loss |
|---|---|---|
| Backtrack | **70.31** | 0.913 |
| Refusal | 60.58 | 1.018 |
| Retrieve | 55.24 | 1.089 |
| Step | 54.17 | 1.184 |
| Sub-query Overlap | 54.35 | 1.015 |

# 4 EXPERIMENTS

We systematically evaluate EVO-RAG on four prominent multi-hop QA benchmarks: HotpotQA, 2WikiMultiHopQA, MuSiQue, and Bamboogle. Our evaluation specifically targets the following three research questions:

## 4.1 RESEARCH QUESTIONS

We evaluate EVO-RAG on four multi-hop QA benchmarks (HotpotQA, 2WikiMultiHopQA, MuSiQue, Bamboogle) and focus on three aspects: (i) overall accuracy/efficiency vs. strong RAG baselines across backbones; (ii) contribution of curriculum/scheduler and the interaction among reward components; (iii) effects of policy objectives (DPO/PPO/GRPO) under identical rollouts/curricula. Results are summarized in Tables 3, 4, 5, 6, and 7.

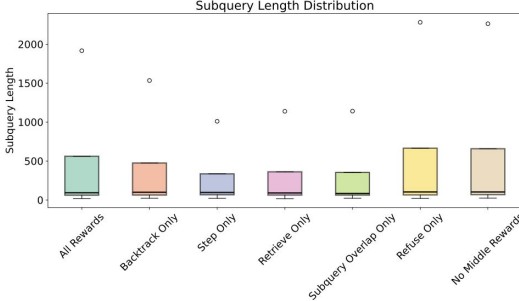 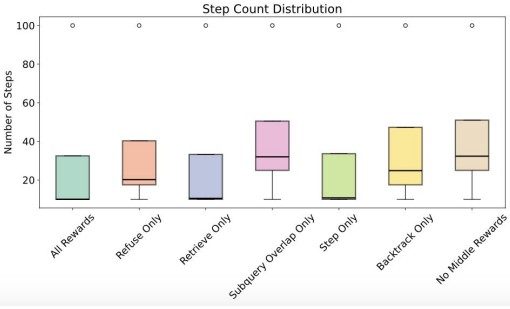

Figure 3: Sub-query length (left) and step count (right) distributions under various reward configurations.

Table 6: Impact of different reward *combinations* on HotpotQA using Qwen2.5-7B-Instruct. Metrics: Exact Match (EM) / F1; Avg. Steps indicates average retrieval length.

| Reward Combination | EM (%) | F1 (%) | Avg. Steps |
|---|---|---|---|
| Baseline (No Reward) | 53.1 | 66.7 | 8.2 |
| Best-2 (Backtrack + AnsCorr) | 56.2 | 70.0 | 11.3 |
| Best-3 (+Overlap) | 56.9 | 70.6 | 10.1 |
| Exploration-Heavy | 55.0 | 69.1 | 13.4 |
| Efficiency-Heavy | 55.4 | 68.8 | 9.0 |
| **Full (All Rewards)** | **57.6** | **71.5** | 10.4 |

Table 7: HotpotQA (LLaMA-3.1-8B) under different objectives.

| Objective | EM (%) | F1 (%) | Avg. Steps |
|---|---|---|---|
| PPO | 55.1 | 69.0 | 11.7 |
| DPO | 55.9 | 69.4 | 11.1 |
| GRPO | **57.4** | **71.2** | **10.2** |

## 4.2 DATASETS AND SETUP

We evaluate EVO-RAG on four multi-hop QA benchmarks. All models are trained using 1,000 queries sampled from HotpotQA. Evaluation is conducted on official validation sets. Answer generation is evaluated using Exact Match (EM) and F1 scores. We intentionally keep training confined to HotpotQA to test cross-dataset generalisation.

**LLM Backbone and Retriever.** We use Llama-3.1-8B-Instruct Touvron et al. (2023), Qwen2.5-7B-Instruction ?, DeepSeek-R1-Distill-Llama-8B Guo et al. (2025) as the agent backbone, paired with RRF-BGE Chen et al. (2024a) retriever (fusion of BM25 Robertson et al. (2009) and BGE embeddings).

## 4.3 RQ1: DO WE IMPROVE OVER STRONG MULTI-HOP RAG BASELINES?

**Main results.** Table 3 compares EVO-RAG against RAG-Gym and IRCoT across three backbones. On **HotpotQA**, EVO-RAG (Qwen2.5-7B-Instruct) reaches **57.6/71.5**, outperforming RAG-Gym (44.1/56.8) by +13.5 EM / +14.7 F1 and IRCoT (45.0/56.2) by +12.6 / +15.3. On **2Wiki**, the best EVO-RAG score (53.2/67.1) exceeds RAG-Gym (50.2/57.9) by +3.0 / +9.2 and IRCoT (45.4/56.8) by +7.8 / +10.3. As an exception, **Bamboogle** favors RAG-Gym (51.2/63.1), while EVO-RAG trained only on HotpotQA scores 45–46 / 58–59 (Table 3), indicating that while EVO-RAG generalizes across standard multi-hop QA, it can underperform on adversarially constructed queries without target-domain tuning.Given Bamboogle's small size (125 items), variance is high; we therefore report bootstrap confidence and treat cross-domain shifts with caution.

**Takeaway (RQ1).** Across three datasets and multiple backbones, EVO-RAG materially improves EM/F1 over strong RAG baselines; the remaining gap on Bamboogle highlights domain-shift sensitivity for adversarial queries.

## 4.4 RQ2: WHAT DESIGN CHOICES MATTER—COMPONENTS AND SCHEDULING?

**(a) Single-reward ablation (component strength).** Time-dynamic scheduling generally helps (DeepSeek/Qwen), with a small exception on Llama-3.1-8B where the fixed two-stage endpoints slightly outperform the in-episode scheduler.

Table 5 trains with one signal at a time and reports internal decision quality (Eval Accuracy/Loss; see Sec. 3.3 and §3.4 for the definition—accuracy of choosing the preferable retrieval action under the preference model). **Backtrack** alone yields the highest internal accuracy (70.31%), indicating that controlled reversibility is a strong driver for robust exploration. **Refusal** ranks second (60.58%), supporting our design to explicitly reward safe abstention when evidence is insufficient.

Pure **Retrieve/Step/Overlap** signals are weaker in isolation, suggesting they are most effective in combination rather than alone.

**Takeaway.** Signals that regulate *control flow* (when to backtrack or refuse) carry disproportionate value; more local efficiency signals need to be paired with them.

**(b) Reward combinations and scheduling (Two-stage vs. Time-dynamic).** *Combinations.* On HotpotQA (Qwen2.5-7B-Instruct), Table 6 shows that moving from Baseline (No Reward) (53.1/66.7, 8.2 steps) to **Best-2** (Backtrack+AnswerCorrectness) already gives +3.1 EM; adding **Overlap** (**Best-3**) both increases EM/F1 (56.9/70.6) and shortens chains (10.1 vs. 11.3). The **Full** configuration (all rewards, time-dynamic) yields the best accuracy **57.6/71.5** at 10.4 steps—longer than Baseline but substantially more accurate, indicating a better accuracy–efficiency trade-off. *Exploration-Heavy* extends chains (13.4 steps) with lower EM (55.0); *Efficiency-Heavy* shortens chains (9.0) but loses EM (55.4).

*Schedules.* Table 4 compares No Reward, Two-stage, and Time-dynamic: **DeepSeek-R1-Distill-Llama-8B**: Time-dynamic ¿ Two-stage ¿ No-Reward (56.8/70.5 vs. 55.0/68.7 vs. 52.6/66.2), i.e., +1.8/+1.8 over Two-stage and +4.2/+4.3 over No-Reward. **Qwen2.5-7B-Instruct**: Time-dynamic likewise wins (57.6/71.5 vs. 55.9/69.5 vs. 53.1/66.7), i.e., +1.7/+2.0 and +4.5/+4.8. **LLaMA-3.1-8B**: Two-stage slightly outperforms Time-dynamic (57.4/71.2 vs. 55.6/69.4), while both clearly beat No-Reward (52.9/66.6).

*Interpretation.* The in-episode scheduler consistently helps (DeepSeek/Qwen), while LLaMA-3.1-8B-Instruct appears to benefit more from fixed stage endpoints. This suggests a backbone–schedule interaction: when the model's search policy is already stable, a smoother decay (Time-dynamic) prevents over-search; otherwise, a stiffer stage separation (Two-stage) may be easier to learn. Figure 2 corroborates the efficiency story: dynamic scheduling suppresses long tails in step counts and sub-query lengths.

**Takeaway (RQ2).** Component-wise, **Backtrack + AnswerCorrectness (+Overlap)** form a strong core; curriculum-wise, the Time-dynamic scheduler is generally superior, with a small exception on LLaMA-3.1-8B-Instruct, where Two-stage wins by a narrow margin.

### 4.5 RQ3: How do policy objectives compare (DPO vs. PPO vs. GRPO)?

**Across datasets/backbones.** Trends are not universal (see Appendix B.1): GRPO tends to help when sibling-action variance is high, DPO is robust when scalarization is brittle or weight tuning is difficult, and PPO can be strong with careful weights.

**Takeaway (RQ3).** On HotpotQA/LLaMA-3.1-8B-Instruct we observe **GRPO** > **DPO** > **PPO** in both accuracy and efficiency see in Table 7; the preferred objective can vary with dataset/backbone and the variance structure of sibling actions.

## 5 Conclusion, Limitations, and Future Work

**Conclusion** We presented EVO-RAG, a two-stage (Discovery→Refinement) agent for multi-hop RAG with seven step-level signals and an in-episode scheduler. Under identical rollouts, DPO/PPO/GRPO experiments show consistent EM/F1 gains while reducing redundancy; ablations highlight the importance of overlap suppression, controlled backtracking, and time-dependent weighting.

**Limitations** Results rely on automatic EM/F1 without human judgments. Reward weights were tuned on HotpotQA and may need retuning elsewhere; refusal validity uses LLM judgments. Actions are prompted rather than learned latents. Compute is modest and performance on adversarial queries (e.g., Bamboogle) is mixed; broader multi-seed statistics are desirable.

**Future Work** We will explore adaptive/meta-learned weights, calibrated uncertainty for stopping/refusal, and latent action policies. Extensions to verification, summarization, and domain retrieval (e.g., legal/patent), plus human evaluation and stronger statistics (multi-seed, paired bootstrap), are planned.

## GENAI USAGE DISCLOSURE

The authors affirm that no part of the paper's text was generated entirely by generative AI tools. Large Language Models (LLMs) were used exclusively for minor grammar editing and formatting suggestions. All code, data annotations, and scientific contributions were created by the authors. The preference model analysis and reward formulation were designed and implemented without GenAI assistance.

## LLM USAGE STATEMENT

We used large language models only for minor grammar edits and formatting suggestions. They did not contribute to problem ideation, experimental design, or writing of scientific content. The authors take full responsibility for all contents.

## ETHICS STATEMENT

This work uses publicly available QA datasets without personal identifying information. We release code to support transparency and reproducibility. The method aims to reduce hallucinations by grounding answers in retrieved evidence. We see no foreseeable harms beyond general concerns of information retrieval bias; we mitigate these by reporting failure cases and enabling refusal when evidence is insufficient.

## REPRODUCIBILITY STATEMENT

We provide training/evaluation scripts, fixed seeds, and configuration files in the supplementary materials; appendix details hyperparameters, compute budget, and dataset preprocessing to facilitate exact replication.

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

## A  REPORTING DETAILS

### A.1  DATASETS AND SPLITS

Benchmarks: HotpotQA (7,404 dev), 2WikiMultiHopQA (12,575), MuSiQue (2,417), and Bamboogle (125); all evaluations use official dev splits.

Unless stated otherwise, we train on a subset of **1,000** HotpotQA training questions sampled uniformly at random, and evaluate on the official validation/dev splits of each dataset. To avoid leakage, we deduplicate question strings and ensure that no evaluation item appears in the training subset.

### A.2  BACKBONES, RETRIEVER, AND ACTION SPACE

Unless noted, the agent backbone is one of **LLaMA-3.1-8B**, **Qwen-2.5-7B**, or **DeepSeek-R1-Distill-Llama-8B** (see Table 3). Retrieval uses **RRF-BGE** (Reciprocal Rank Fusion of BM25 and BGE embeddings). At hop $t$, the environment returns the top-$k$ passages (constant $k$ across runs). The discrete action set is {SEARCH, BACKTRACK, ANSWER, REFUSE} with an episode cap $T_{\max}=20$ steps.

### A.3  REWARD INSTRUMENTATION AND SCHEDULER

We keep the final-answer reward (EM/F1) and *decompose* process supervision into seven step-level signals (Sec. 3.3): Retrieval Bonus, Retrieval Action Penalty, Sub-query Overlap Penalty, Backtrack Penalty, Refusal Reward, Step Penalty, and Answer Correctness. Signals are either rule-based (environment-computable) or LLM-verified (only for semantics-heavy cases, e.g., justified refusal). We schedule step-level weights within each episode using a progress ratio

$$p(t) = \frac{t-1}{T_{\max}-1} \in [0,1], \qquad w_k(t) = (1-p(t))\,w_k^{\text{early}} + p(t)\,w_k^{\text{late}}.$$

with *stage-dependent endpoints* (Table 2). Thus, Discovery uses exploration-leaning endpoints and Refinement uses efficiency/accuracy-leaning endpoints, while the *actual* per-step weights continue to evolve within every episode.

**Training-time stage switch.** We switch from Discovery to Refinement when the *composite scalarized return* on a held-out dev split plateaus. Let $R(\tau)=\sum_t \sum_k w_k(t)\, r_t^{(k)}$. We track the exponential moving average $\widehat{J}$ of per-episode returns; when the improvement over the best running $\widehat{J}$ is $< \varepsilon$ for $P$ consecutive checkpoints (patience), we trigger the stage switch. Intuitively, once exploration no longer increases composite return, endpoints are shifted toward precision and efficiency.

### A.4  PREFERENCE MODEL AND PAIR CONSTRUCTION

From rollouts labeled with step-level signals $\{r_t^{(k)}\}_{k=1}^7$ and weights $\{w_k(t)\}$, we form *trajectory-prefix* preference pairs $(x^+, x^-)$ using the weighted return $\sum_t \sum_k w_k(t) r_t^{(k)}$. A shared encoder with seven linear heads $\{f_\phi^{(k)}\}_{k=1}^7$ predicts aspect-wise scores and is trained with a head-wise logistic loss:

$$\mathcal{L}_{\text{RM}} \;=\; -\tfrac{1}{7} \sum_{k=1}^7 \log \sigma\big(f_\phi^{(k)}(x^+) - f_\phi^{(k)}(x^-)\big).$$

Sibling candidates at the same step are used to increase pair diversity; we keep positive/negative balance close to 1:1 by down-sampling the majority side.

### A.5  POLICY OBJECTIVES AND ADVANTAGE ESTIMATION

We benchmark three objectives under identical rollouts and curricula:

- **DPO** optimizes preferences directly, $\mathcal{L}_{\text{DPO}} = -\log \sigma\big(\beta_{\text{dpo}}[\log \pi_\theta(x^+) - \log \pi_\theta(x^-)]\big)$.

- **PPO** uses scalarized step rewards $\tilde{r}_t = \sum_k w_k(t) r_t^{(k)}$ with GAE advantages and clipped updates.

- **GRPO** replaces $A_t$ with group-normalized advantages across sibling candidates: $\hat{A}_t^{(i)} = (\tilde{r}_t^{(i)} - \mu_t)/(\sigma_t + \varepsilon)$.

Hyperparameters for each objective are held constant across backbones; exact configs are released with the code.

### A.6  TRAINING PROTOCOL AND COMPUTE

Unless stated otherwise, we report the mean over **3 random seeds** (same seeds across all methods and backbones). We use a single high-memory GPU and mixed-precision training. Each run alternates (i) rollout collection with the current scheduler and (ii) policy updates (Algorithm 1). We save checkpoints at fixed intervals and select the best dev EM for reporting.

### A.7  EVALUATION PROTOCOL

All metrics use the official evaluation scripts of each dataset. HotpotQA, 2WikiMultiHopQA, and MuSiQue are scored by EM and F1; Bamboogle by EM/F1 following prior work. We also track *Avg. Steps* (average retrieval depth) to quantify efficiency. Unless explicitly noted, no target-domain fine-tuning is performed (Table 3, [†]).

### A.8  STATISTICAL TESTING AND UNCERTAINTY

For tables that aggregate over multiple seeds, we report mean±std. For pairwise method comparisons on EM/F1, we run a *paired bootstrap* with 10,000 resamples over per-question predictions and mark differences significant at $p < 0.05$. When box plots are shown (e.g., Fig. 3), whiskers mark 5th–95th percentiles.

### A.9  ABLATIONS AND CONTROLS

We include: (i) **single-signal training** (Table 5), (ii) **reward-combination** studies (Table 6), and (iii) **schedule variants** (No Reward, fixed Two-stage, Time-dynamic; Table 4). Unless specified, all other settings are unchanged.

### A.10  REPRODUCIBILITY ARTIFACTS

We release scripts to (a) materialize the training subset, (b) reproduce all rollouts, (c) run DPO/PPO/GRPO with the same scheduler, and (d) evaluate and bootstrap metrics. All random seeds, checkpoint hashes, and configuration files are included.

Table 8: HotpotQA with LLaMA-3.1-8B-Instruct under different objectives. Mean over 3 seeds.

| Method | EM (%) | F1 (%) | Avg. Steps |
|---|---|---|---|
| PPO (scalarized reward) | 55.1 | 69.0 | 11.7 |
| DPO (multi-preference) | 55.9 | 69.4 | 11.1 |
| GRPO (group-normalized) | **57.4** | **71.2** | **10.2** |

Table 9: Pilot study of adaptive reward weight tuning on HotpotQA (LLaMA-3.1-8B).

| Method | EM (%) | F1 (%) | Avg. Steps |
|---|---|---|---|
| Manual schedule (main paper) | 57.4 | 71.2 | 10.2 |
| Bayesian optimization (BO) | 57.8 | 71.5 | 10.1 |
| Bandit-based (UCB1) | 57.6 | 71.3 | 10.4 |

# B  ADDITIONAL RESULTS AND ANALYSES

## B.1  EFFECT OF POLICY OBJECTIVE (DPO VS. PPO VS. GRPO)

## B.2  ADAPTIVE REWARD WEIGHT TUNING (PILOT STUDY)

**Discussion.** Adaptive methods yield comparable or slightly better accuracy than the manual schedule. BO converged to weights close to our hand-tuned configuration with marginal EM/F1 gains; UCB1 adapted weights online without manual intervention. These confirm the feasibility of adaptive tuning for robustness and cross-domain transfer (e.g., MuSiQue, Bamboogle).

# C  PARAMETERS USED IN VERL

Table 10: VERL training hyperparameters by objective (shared unless noted).

| Hyperparameter | PPO | GRPO |
|---|---|---|
| Critic model path | Qwen/Qwen2.5-0.5B-Instruct | – |
| LR (actor / critic) | 1e−6 / 1e−5 | 1e−6 / – |
| Train batch size (episodes) | 128 | 16 |
| PPO mini-batch size | 64 | 8 |
| Micro-batch / GPU | 4 | 2 |
| Max prompt / response length | 2048 / 256 | 1330 / 256 |
| KL coef ($\lambda_{\mathrm{KL}}$) | 0.001 | 0.0 |
| Adv estimator | PPO (GAE) | GRPO |
| TP size (vLLM) | 1 | 2 |
| vLLM gpu_mem util | 0.4 | 0.4 |
| Rollout n (per prompt) | – | 1 |
| Critic warmup | – | 0 |
| Epochs / save freq / test freq | 15 / 10 / 10 | 15 / 10 / 10 |
| GPUs (per node) | 1 | 2 |
| Seeds | 3 | |

# D  TRAINING LOOP (FOR COMPLETENESS)

---

**Algorithm 1** EVO-RAG training loop

---

1: Initialize policy $\pi_\theta$ and preference model $f_\phi$
2: **for** stage $\in \{\text{Discovery}, \text{Refinement}\}$ **do**
3:     **for** $m = 1$ **to** $M$ episodes **do**
4:         Roll out with dynamic weights $w_k(t)$; collect trajectories $\tau$ and sibling pairs $(x^+, x^-)$
5:     **end for**
6:     **Update** $f_\phi$ by minimizing $\mathcal{L}_{\text{RM}}$ on collected $(x^+, x^-)$
7:     **Update** $\pi_\theta$ by minimizing $\mathcal{L}_\mathcal{O}$ with $\mathcal{O} \in \{\text{DPO}, \text{PPO}, \text{GRPO}\}$
8: **end for**

---

# E  CASE STUDIES

Table 11: Compact traces under different reward schedules. "Dup." = near-duplicate; ◊ marks the timestep preferred by the reward model. Correct answers are **bold**; wrong ones in red.

|  | **Baseline (No Reward)** | **Two-stage (Fixed)** | **Time-dynamic (EVO-RAG)** |
|---|---|---|---|
| **Q1: "In which year was the monarch who issued the 1925 Birthday Honours born?"** | | | |
| Steps | 1 | 2 | 2 |
| Main hops | $q_1$: direct ask → noisy list | $q_1$: same; $q_2$: ask birth (Dup.◊) | $q_1$: identify monarch; $q_2$: ask birth |
| Outcome | 1867 | **1865** | **1865** |
| **Q2: "Which U.S. state contains the launch site of Mars Pathfinder?"** | | | |
| Steps | 1 | 6 | 2 |
| Main hops | $q_1$: "launch site state" → LC-17 | $q_{2..5}$: "launch pad" paraphrases (Dup.) | $q_1$: site → Cape; $q_2$: "Which state?" |
| Outcome | California | **Florida** | **Florida** |
| **Q3: "Where was the 2021 Hugo Award ceremony hosted?" (unanswerable)** | | | |
| Steps | 1 | 13 | 4 |
| Main hops | $q_1$: host city; no citation | many paraphrases (Dup.) | tried a few; stopped timely |
| Outcome | London | Dublin | *REFUSE* (✓) |