# OpenReview forum: "EVO-RAG: Evolving Retrieval-Augmented Agents for Efficient Multi-Hop Query Optimization"
_ICLR.cc/2026/Conference — ICLR 2026 Conference Withdrawn Submission_

### Official Review · Reviewer_2cS7 · 2025-10-20

**Soundness:** 1
**Presentation:** 1
**Contribution:** 1
**Rating:** 2
**Confidence:** 4

**Summary:**

This paper proposes a phase-aware framework for RAG tasks with 7 step-level rewards and a time-based scheduler. The key idea is to train a reward model with fine-grained seven step-level signals and use it to facilitate the entire RAG pipeline. Specifically, the authors propose a multi-head preference model and train the policy model with DPO, PPO, and GRPO. Experiments on four multi-hop QA datasets demonstrate that EVO-RAG achieves significant improvements over existing baselines.

**Strengths:**

1. The proposed seven step-level signals for RAG tasks are novel and could potentially facilitate process supervision in RL training.

2. The authors investigate the impact of different training paradigms (DPO, PPO, and GRPO), providing insights into their relative effectiveness.

3. The paper evaluates on multiple multi-hop QA datasets, demonstrating the generalizability of the approach.

**Weaknesses:**

1. The paper appears to be an early draft rather than a submission ready for ICLR. Specific issues include:
  * Figure 1 is confusing and inconsistent with the main text. For example, `Refusal Reward`, `Backtrakc Penalty`, and `Retrieval Action Penalty` are classified as LLM-based rewards in Figure 1, but Table 1 only lists `Refusal Reward` under LLM-based Rewards.
  * It would be helpful if the authors could clarify the purpose, motivation, and the detailed cases of each action (such as Backtrack, Refuse) and step-level signals. In Appendix E (Case Studies section), the authors seem to only provide a table without any explanatory text.
  * The entire Method section appears to lacks motivation and rationale for design choices, making it difficult to understand the proposed approach.
  * There are some missing references in Line 404 and 446.
  * The discussion in Section 4.5 is insufficient. The authors refer readers to Appendix B.1 for the trends of DPO, PPO, and GRPO, but Appendix B.1 is empty.

2. The paper lacks ablation studies to validate the necessity of each of the seven proposed signals. Without understanding which components actually contribute to performance, the results are not convincing.

3. The paper does not compare with recent state-of-the-art RAG methods, making it difficult to assess the actual improvements.

**Questions:**

1. Could you clarify the inconsistencies between Figure 1 and Table 1?
2. What is the main technical novelty beyond combining existing methods?
3. Why is Appendix B.1 empty when referenced in the main text?

---

### Official Review · Reviewer_XYg1 · 2025-10-28

**Soundness:** 2
**Presentation:** 2
**Contribution:** 2
**Rating:** 4
**Confidence:** 3

**Summary:**

The paper “EVO-RAG: Evolving Retrieval-Augmented Agents for Efficient Multi-Hop Query Optimization” proposes an RL-based approach to improve multi-hop Retrieval-Augmented Generation (RAG). The method introduces a two-stage curriculum (Discovery → Refinement) combined with seven interpretable step-level rewards and a time-based scheduler that gradually shifts focus from exploration to efficiency. The authors benchmark different optimization objectives (DPO, PPO, GRPO) on several QA datasets, showing moderate improvements in EM/F1 and reduced redundant hops compared to RAG-Gym and IRCoT baselines.

**Strengths:**

Originality

 Offers a structured framework for multi-hop retrieval training using a two-phase curriculum and time-scheduled step-level rewards, which helps formalize the retrieval reasoning process.
 Introduces an interpretable multi-head preference model that decomposes training signals into meaningful sub-rewards.
 Provides a rare controlled comparison of DPO, PPO, and GRPO objectives under identical rollouts.

Quality

 The experiments are well-organized and include ablations, scheduler variants (Two-stage vs. Time-dynamic), and multiple model backbones.
 The methodology is transparent and reproducible, with detailed pseudocode, hyperparameters, and open-source code.
 The empirical gains, though moderate, are consistent across several benchmarks, demonstrating the framework’s practical effectiveness.

Clarity

 Figures and tables are informative and help clarify the method’s design (especially Fig. 1–2).
 Appendices contain comprehensive implementation details, supporting reproducibility.
 The overall paper structure is logical and easy to follow.

Significance

 Addresses a key challenge in retrieval-augmented reasoning—balancing exploration with efficiency.
 The modular reward design may influence future RLHF and process-level supervision approaches for retrieval-based agents.

**Weaknesses:**

1. Incremental Contribution

    The main technical elements—multi-signal rewards, simple time scheduling, and a two-stage curriculum—are well-known ideas in RLHF and curriculum learning. The novelty lies mostly in combining them rather than introducing fundamentally new concepts.
    The “evolving” mechanism is manually designed rather than adaptively learned, limiting the originality of the contribution.

2. Writing and Presentation Issues

    There are some issues with the writing and formatting that detract from the paper’s polish, such as a missing or broken citation (e.g., “?” on line 404) and minor LaTeX inconsistencies.
    Certain explanations could be clearer—for instance, how the scheduler interacts with the two-stage curriculum or how preference scores are used in practice.
    These are not major flaws but suggest the paper would benefit from additional editing and proofreading before publication.

3. Limited Theoretical or Conceptual Depth

    The paper provides little theoretical analysis of how the reward structure ensures convergence or stability.
    Reward weights and schedules are fully manual (Table 2), and there is no investigation into learned or adaptive scheduling strategies.

4. Narrow Experimental Scope

    The training set is restricted to 1,000 HotpotQA samples, which weakens the generalization claims.
    Performance gains are modest and inconsistent; on Bamboogle, EVO-RAG performs worse than simpler baselines.
    No human evaluation or detailed error analysis is provided.

5. Overstated Claims

    Terms like “evolving” and “phase-aware agents” imply adaptive behavior not supported by the implementation.
    The contribution could be more accurately described as an integrated reward shaping framework rather than a fundamentally new learning paradigm.

6. Incomplete Literature and Citation Coverage

    The paper could more clearly position itself relative to recent process-level RLHF methods (e.g., Step-DPO, RLAIF, Self-Rewarding RAG).
    A few citation formatting errors further suggest editorial oversights.

**Questions:**

1. How sensitive are the results to manually chosen reward weights and schedule parameters (Table 2)?
2. Could adaptive or uncertainty-driven scheduling outperform linear decay?
3. Would fewer composite rewards (e.g., combining efficiency and overlap penalties) suffice?
4. How does the approach compare with recent step-level preference optimization methods?
5. Will fixing the missing citation and improving exposition change clarity of the related work section?

---

### Official Review · Reviewer_2bzn · 2025-10-31

**Soundness:** 3
**Presentation:** 3
**Contribution:** 3
**Rating:** 2
**Confidence:** 4

**Summary:**

The paper introduces a phase-aware reinforcement learning framework EVO-RAG to improve retrieval-augmented generation (RAG) for multi-hop question answering. EVO-RAG divides reasoning into two stages—Discovery and Refinement—and uses seven interpretable step-level rewards (retrieval hit/miss, action penalty, sub-query overlap, backtrack, justified refusal, step cost, and answer correctness). An in-episode time scheduler dynamically shifts reward weights from exploration toward efficiency and accuracy as reasoning progresses. The system integrates a multi-head preference model and compares DPO, PPO, and GRPO objectives under identical rollouts.

**Strengths:**

1. Introduces fine-grained, time-aware reward scheduling for RAG, improving interpretability and control.
2. Benchmarked on multiple QA datasets and compared fairly across objectives, and achieve consistent EM/F1 improvements and reduced redundancy.

**Weaknesses:**

1. As the author indicates, the performance on adversarial or domain-shifted datasets (e.g., Bamboogle) is not impressive. My main concern is the limited generalization of this method.
2. The method depends on heuristic reward tuning, and the weights require manual adjustment per dataset. Is there any way to make it more efficient? The discovery to refinement transition is based on a plateau criterion in dev-set reward, which may not generalize across domains or dynamically changing query complexities. The two-stage assumption enforces a simplified linear early explore to late refine trajectory that might not fit tasks requiring intermittent exploration (e.g., returning to earlier evidence)
3. The refusal reward depends on an external LLM judge to verify whether a question is unanswerable. Is there any human evaluation to measure the consistency between LLM judge and human evaluation? In addition, only automatic EM/F1 metrics are reported, which also could be evaluated by human. As the performance of LLMs, human evaluation / alignment is very important.

**Questions:**

See weaknesses.

---

### Official Review · Reviewer_52re · 2025-11-01

**Soundness:** 2
**Presentation:** 2
**Contribution:** 4
**Rating:** 6
**Confidence:** 4

**Summary:**

The paper introduces EVO-RAG for multi-hop retrieval-augmented generation that integrates a two-stage curriculum (Discovery to Refinement) and seven interpretable step-level rewards. Unlike prior approaches that rely on static or episode-level supervision, EVO-RAG introduces a time-based scheduler that dynamically shifts reward weights within an episode, decreasing exploration incentives and increasing efficiency/accuracy pressure as reasoning progresses. EVO-RAG is trained using a multi-head preference model and supports DPO, PPO, and GRPO under identical rollouts. Authors use standard datasets - HotPotQA, MuSiQue, 2WikiMHQA, Bamboogle for comparison with baselines.

**Strengths:**

1. On a high level: the paper addresses a genuine gap that current RAG systems often lack phase-awareness, leading to over-exploration early or premature stopping late.

2. The proposed in-episode time based reward is intuitive and the two-stage curriculum offers a structured way to balance exploration and exploitation.

3. The step level rewards are reasonable and interpretable.

4. The discovery that Backtrack + Answer Correctness (+Overlap) are the most valuable signals is well-supported.

5. The study also probes backbone-specific scheduler effects (e.g., LLaMA benefits slightly more from Two-Stage than Time-Dynamic).

6. EVO-RAG achieves consistent improvements across 3 major multi-hop QA benchmarks, using 8B-class backbones trained on only 1,000 HotpotQA examples.

7. The paper is very well-organized, with exhaustive appendices detailing hyperparameters, seeds, scripts, and configurations.

**Weaknesses:**

1. The baseline comparison (RAG-Gym, IRCoT) omits several recent SOTA RAG methods, such as: CoRAG (Wang et al), R1-Searcher (Song et al), FrugalRAG (Java el al), O2-Searcher (Mei et al), SimpleDeepSearcher (Sun et al), etc. For example, FrugalRAG also uses an explore-exploit strategy to train the policy with 1000 examples. SimpleDeepSearcher uses 871 examples to train their model. Comparing against these would more convincingly position EVO-RAG within current research trends.

2. While the paper includes four datasets, only 2-4 hop QA benchmarks are used (HotpotQA, 2Wiki, Musique, Bamboogle). These are relatively shallow multi-hop tasks by current standards.

3. The paper would benefit from better presenation of results. Specifically, te F1/EM metrics in Table 3 are stacked (e.g., 57.8 / 71.4), but presented as single cells. A clearer table format (split columns) would aid readability and facilitate significance interpretation.

4. The LLM used for REFUSE validation (step-level reward for justified refusal) is not explicitly named. Is it the same backbone (e.g., LLaMA-8B) or an external verifier model?

5. The computational overhead of seven concurrent rewards and a preference model is not quantified. While EVO-RAG claims lightweight inference, the training process is computationally intensive. Seven concurrent reward signals, a multi-head preference model, and separate optimization under PPO/DPO/GRPO all add non-trivial cost. In particular, the LLM-based refusal verification and preference-pair construction could make scaling to larger datasets or longer reasoning chains expensive. Quantitative reporting of compute cost, GPU hours, or per-episode latency would strengthen transparency.

6. Cross-dataset generalization is encouraging to some extend but limited to wikipedia-style datasetss (notably weak on Bamboogle). The authors do acknowledge this in their paper.

Overall: I would recommend accepting this paper given my concerns regarding limited evaluations and metric presentation are resolved.

**Questions:**

Please see weaknesses.

---

### Note · Authors · 2025-11-12

I have read and agree with the venue's withdrawal policy on behalf of myself and my co-authors.